# Applying Ternion Stream DCNN for Real-Time Vehicle Re-Identification and Tracking across Multiple Non-Overlapping Cameras

**DOI:** 10.3390/s22239274

**Published:** 2022-11-28

**Authors:** Lesole Kalake, Wanggen Wan, Yanqiu Dong

**Affiliations:** Institute of Smart City, School of Communications and Information Engineering, Shanghai University, Shanghai 200444, China

**Keywords:** vehicle tracking, re-identification, three stream, deep convolutional neural network, non-overlapping multiple cameras

## Abstract

The increase in security threats and a huge demand for smart transportation applications for vehicle identification and tracking with multiple non-overlapping cameras have gained a lot of attention. Moreover, extracting meaningful and semantic vehicle information has become an adventurous task, with frameworks deployed on different domains to scan features independently. Furthermore, approach identification and tracking processes have largely relied on one or two vehicle characteristics. They have managed to achieve a high detection quality rate and accuracy using Inception ResNet and pre-trained models but have had limitations on handling moving vehicle classes and were not suitable for real-time tracking. Additionally, the complexity and diverse characteristics of vehicles made the algorithms impossible to efficiently distinguish and match vehicle tracklets across non-overlapping cameras. Therefore, to disambiguate these features, we propose to implement a Ternion stream deep convolutional neural network (TSDCNN) over non-overlapping cameras and combine all key vehicle features such as shape, license plate number, and optical character recognition (OCR). Then jointly investigate the strategic analysis of visual vehicle information to find and identify vehicles in multiple non-overlapping views of algorithms. As a result, the proposed algorithm improved the recognition quality rate and recorded a remarkable overall performance, outperforming the current online state-of-the-art paradigm by 0.28% and 1.70%, respectively, on vehicle rear view (VRV) and Veri776 datasets.

## 1. Introduction

Traffic monitoring is an indispensable tool used for collecting statistics to enable better design and control of transport infrastructure [1,2,3]. As a result, many applications have emerged to improve traffic management, focusing only on vehicle counting in urban streets [4]. However, plain vehicle counting was proven not to be sufficient for locating and distinguishing between vehicle types or models [5]. Then vehicle localization and identification [6] have become an area of interest in the computer vision community for solving vehicle-related criminal activities such as theft in urban areas [6]. Additionally, numerous research projects were conducted to solve various environmental challenges in vehicle detection, re-identification, and tracking across multiple camera views [5,7]. However, most of these proposed algorithms are observed offline and are not ideal for real-time tracking [8]. Moreover, they make it difficult for human observers [9] to remember and efficiently distinguish between a wide variety of vehicle makes and models [10]. It became an arduous task for a human being to monitor dozens of screens for incoming and outgoing vehicle models [11]. Therefore, in an attempt to resolve this issue, refs. [12,13,14,15,16] proposed algorithms to distinguish vehicles based on shapes, size, traveling speed, and distance from camera views. However, the algorithms disregarded the vehicle’s visual information and resulted in poor overall performance. Then refs. [17,18,19] extended the vehicle visual information using Faster-RCNN on camera videos and extracted more vehicle attributes for recognition and classification categories. However, these algorithms struggled to detect and retrieve vehicle license plates and characteristics in various classes with similar shapes and designs. Additionally, their tracking performance deteriorated on low-resolution plates and with illumination variations.

Using the existing infrastructure, Tian et al. [20] extracted and collected vehicle signature profiles from inductive loop detectors [21], weight-in-motion devices [22], and micro-loop sensors [23] to analyze vehicle behavior and pattern trends. However, the signature-based technique was complex and relied on a complex data model, resulting in significant processing resources. Additionally, Koetsier et al. [24] suggested the implementation of DCNNs for fine-grained vehicle categorization and model verification. Then, Khorramshahi et al. [25] investigated the benefits of integrating low-level and high-level semantic characteristics for a vehicle search based on the fusion process for appearance features such as texture (car plates), color, and DCNN-learned semantic attributes. However, in some frames, poor quality detection limited the extraction of appearance features for vehicle instances. As a result, it limited the ability of these algorithms to distinguish between vehicles of the same model and recognize the license plates.

In an attempt to address vehicle distinguishing problems, Subhalakhsmi and Siva [26] developed an automatic license plate recognition system that tracks vehicles based on license plate recognition and color. Their algorithm consisted of a hybrid decision tree and an SVM classifier with a fifth-degree homogeneous polynomial kernel. It performed the license plate localization by identifying the position of the license plates in the original images. However, Zhang et al. [27] argued that vehicle identification could not be solely based on license plate information, particularly in the presence of occlusions and plate duplicates. Moreover, they emphasized that even the color and license plate recognition were unfair enough to distinguish between vehicle similarities in appearance. In support of these findings, Hashem et al. [11] concluded in their survey that most algorithms used to recognize license plates and detect vehicles were hampered by distortions, illumination variations, and occlusions. As a result, DCNN was voted the best technique to enhance color-based vehicle recognition and tracking applications.

Using these discoveries, Dehghan et al. [28] developed an application to re-identify vehicles from video images based on make, model, and color classification. Their algorithm, however, failed to distinguish the similarities between vehicles. This left a noticeable research gap for more investigations to be conducted on the vehicle model classification. Therefore, these developments formulated a clear research gap in this area. Hence, Biglari et al. [29] crosschecked the vehicle make and model with the license plate by combining the global and local information. Then refs. [30,31] introduced a dual car recognition framework that relies on the analysis of the vehicle’s external features. The first framework is used to evaluate the shape of the vehicle’s rear by exploring the dimensions and edges, and the second focuses on the features computed from the vehicle’s rear lights. Finally, both of these frameworks were combined to detect a moving vehicle’s make and model based on integrated CNN. Inspired by this architecture, Ke et al. [32] continued with an investigation on how to fine-grain vehicle recognition using DCNN. Their algorithm localizes the vehicle and corresponding parts using the RCNNs. It further aggregated the features from a set of pre-trained CNNs to train the SVM classifier. However, the above-mentioned approaches continued to experience challenges in distinguishing similarities between vehicles across multiple cameras in real-world practical problems.

Komolovaite et al. [33] then attempted to address the challenges in vehicle similarity appearance using the transfer-learning framework. They transferred the learned knowledge from pre-trained AlexNet to learn and classify the vehicles’ front and rear views. Furthermore, to evaluate their algorithm, they developed a similar network to AlexNet with frozen weights for feature extraction on network+SVM. The proposed transfer-learning framework outperformed the newly built network and recorded noticeable accuracy. However, the framework had trouble handling similarities in shapes, sizes, and colors of the vehicles, and this resulted in poor detection.

In conceptualizing the multiple steam for multi-task handling in the computer vision community, refs. [34,35] illustrated how the three streams CNN with multi-task can be the ensemble to learn and classify 3D objects. This inspired [36,37], who proposed multi-level feature extraction DCNN for vehicle Re-ID across non-overlapping cameras. Their algorithm illustrated robustness in combining the small image patches but had difficulty recognizing the far, blurry, and out-of-range vehicle images. Moreover, it could not differentiate the vehicles of the same make, model, and color [38] with different plates.

Inspired by these discoveries, we proposed the Ternion streams DCNN to detect, track, and Re-ID vehicles. Our algorithm uses three streams of DCNN to provide an accurate estimation direction for the number of moving vehicles based on non-overlapping camera views. We investigated and used each stream independently to extract vehicle characteristics such as shapes and plates with an integrated OCR stream for low character resolutions, then, fuse these attributes from the streams to form a complete vehicle surveillance recognition and tracking system. We further used distance descriptors and vectors to measure the vehicles’ similarities. This paper is organized as follows: in Section 1 we give an introduction and background; Section 2 describes the Ternion streams DCNN framework; Section 3 lists experiment materials and parameter settings; Section 4 presents Results; in Section 5, we discuss and analyze the results; and then finally we draw up the paper conclusions in Section 6.

## 2. Proposed Ternion Stream DCNNs for Real-Time Vehicle Tracking (VT)

The main task is to track and re-identify the target across these multiple cameras [19,20,21]. We, therefore, designed our algorithm to detect, track and re-identify the vehicles across several non-overlapping cameras based on the Ternion stream DCNN (TSDCNN) framework. We implemented the proposed algorithm using the dataset that contains different shapes of vehicles [23] and different illumination conditions. The algorithm relies on three streams of the DCNN, which are independently used for extracting vehicle characteristics like shapes and plates. The detection of plates was buttressed [24,26] with the integration of the OCR stream, which has proven to cater to illegible characteristics on damaged plates and low resolutions. According to our best knowledge, there is no similar proposed algorithm for real-time object tracking across multiple non-overlapping cameras. The TSDCNN architecture and working procedure are presented in Figure 1. The model is trained using datasets with multiple vehicle types’ videos. However, the input to the first stream is the sequences of RGB frames’ 96 × 96 pixel cropped images for an application detector to perform detection, Re-ID processes, and searches using two phases. In the first phase, the model extracts rear vehicle shapes and searches if a similar vehicle from the video frame images has already appeared over the camera network. This is performed solely based on vehicle shape appearance; whereas, in the second step, we extract the vehicle with plates on a small region and feed them onto the plate stream DCNN. However, the detection and reading of plates on low resolutions became more challenging in the intra-class similarities, viewpoint changes, and inconsistent environmental conditions. We, therefore, introduced and integrated the OCR stream to minimize these problems. Then, we agglutinate the streams to obtain more comprehensive vehicle information for distinguishing, recognizing, and associating the vehicle’s tracklets across non-overlapping cameras. Additionally, we assumed that $V=\left\{V_{1\ldots }V_{n}\right\}C_{i}R^{p}$ should represent the collection of n vehicles to be recognized and classified. Furthermore, we merged the features’ distance descriptor vectors and independently shared the weights within each stream. We then applied the Euclidean method to calculate the similarity based on the distance between the license plates of the query image and the multiple cameras’ video frame gallery images.

Additionally, we combined the three streams’ extracted features and computed the similarities between vehicle images. Finally, we added these to the convolutional fully connected layers and activated the softmax function for vehicle matching and classifications.

By implementing this strategy, our algorithm staunchly recognizes and classifies the vehicle $\left\{V_{i}\epsilon V\right\}$ based on shapes, plates, and OCR. Furthermore, it located the vehicle in every class $\left\{C_{i}\epsilon C\right\}$. As a result, we rewrite the formulas as the following:(1)$$V=\sum_{i}^{n}V_{i}C_{i}R^{p}$$
where p denotes vehicle features, $V=\left\{V_{1\ldots }V_{n}\right\}$ represents a subset of the vehicle sample, and $C=\left\{C_{1\ldots }C_{n}\right\}$ denotes the sunset of the vehicle class sample.

### 2.1. Data Collection and Preparation Process

We used public datasets (VeRi 776 and vehicle rear view (VRV)), which are a collection of video sequences captured with non-overlapping cameras. The videos $V_{vid}=\left\{V_{vid_{1}}\ldots V_{vid_{N}}\right\}$ are the input source and have been converted to frames $f_{r}=\left\{f_{r_{1}}\ldots f_{r_{N}}\right\}$, which are further subdivided into images $Img=\left\{Img_{1}\ldots Img_{N}\right\}$ (see Figure 2). However, the images extracted from the sequential video frames are treated interchangeably as simple input sources that are fed to the Ternion stream DCNN. Then the special vehicle features such as shape, license plate, and low-resolution characters are recognized simultaneously and extracted from the input. Additionally, morphological operations and segmentation techniques are used to remove background noise. To improve vehicle detection and readability for small areas, the OCR stream has been added as a third stream to the two streams, i.e., shapes and plate streams. The Ternion streams DCNN is then used to independently extract the unique vehicle characteristics. However, to create an efficient framework, we shared the parameters among the stream networks and considered implementing the ROI to generate appropriate bounding frames. As a result, the vehicle tracking centroids and Euclidean distance methods were used in the calculation of the distance and movement of the vehicle from frame to frame. Moreover, we calculated the similarities between frames and estimated the loss by feeding the last layers of the framework to the contrastive loss function as the following:(2)$$C_{loss}=\left(1-Y\right)\frac{1}{2}{\left(D_{w}\right)}^{2}+\left(Y\right)\frac{1}{2}{\left\{\max\left(0,m-D_{w}\right)\right\}}^{2};\;\;0\leq Y\leq 1,\;m>0,\;0\leq max\leq m-D_{w}$$
where $D_{w}$ denotes the Euclidean distance between the outputs of the Ternion stream DCNN framework.

Thus, we further expressed $D_{w}$ as:(3)$$D_{w}=\sqrt[{2}]{\left\{G_{w}\cdot X_{1}-G_{w}\cdot X_{2}\right\}}$$
where, $G_{w}$ denotes the output of the framework, m represents the margin value, and *Y* values indicate if inputs are from the same class.

### 2.2. Appearance Features’ Learning and Handling

Currently, existing vehicle Re-ID methods typically only extract the global appearance characteristics. However, since vehicles of the same make and model have a similar global appearance, extracting only global features for vehicle re-identification makes it difficult to distinguish between them. Consequently, both global and local attributes are crucial for improving the feature representation, discrimination, and robustness of vehicle algorithms. We, therefore, propose using both global and local inputs to build more discriminatory representations of the vehicles. Additionally, we introduce the three-stream DCNN frameworks, where the built-in OCR serves as a third stream to process the low-resolution and illegible plate characteristics. The proposed algorithm inputs the images from sequential frames and divides them into regions, which are fed to the framework for predicting bounding boxes and ROI likelihood values. However, each stream is applied simultaneously to extract vehicle features and to generate discriminators independently with separate weights. Additionally, we flattened the features from the ROI pooling layer into a vector and fed the final fully connected layers into the frameworks. Finally, the feature vectors of the streams were concatenated to form significant unique information for vehicle detection, tracking, Re-ID, and matching.

### 2.3. License Plate-Based Vehicle Re-ID

Depending on the distance from the cameras, multiple plate detection and recognition are frequently available in chaotic conditions. Therefore, due to illumination, viewing angle, and occlusion, this makes it challenging. Additionally, it has been proven to facilitate feature isolation during the segmentation process; to tackle these issues, several algorithms have emerged and rely on template-matching implementation. The template matching technique, however, was ineffective and did not offer a durable answer. However, to validate and match license plates, we suggest using the plate stream neural network, and from the pairings of license plates, the stream is intended to generate robust discriminative feature vectors. Moreover, to make the plate symbols and characters easier to read, it is also combined with the parallel neural network OCR stream. The two neural network streams are continuously and consistently trained to match the output pixel images. This improved our method as it became possible to design features resistant to geometric distortions in the query image and to learn the best shift-invariant local feature detectors. To ensure that the distance between license plate pairs of the same vehicle is small and the distance between license plate pairs of different vehicles is large, we also calculated the Euclidean distance between the feature vectors.

This problem is presented and solved mathematically, as follows: supposing $\varrho_{1}$ and $\varrho_{2}$ are input pairs of vehicle ν license plates, ς is a binary label of the pair Θ. The Euclidean distance is then calculated as follows:(4)$$E_{w}(\varrho_{1},\;\varrho_{2})\;=\;∥\zeta_{w}\left(\varrho_{1}\right)-\zeta_{w}\varrho_{2}∥\;\mathrm{if}\;\left(\varrho_{1},\varrho_{2}\right)\;\epsilon \;\nu ;\;\mathrm{then}\;\{\begin{array}{c} \theta =1\\ otherwise\;0\; \end{array}$$
where w denotes the weights of the convolutional neural network, and $\zeta_{w}\left(\varrho_{1}\right)$ and $\zeta_{w}\varrho_{2}$) represent features extracted from $\varrho_{1}$ and $\varrho_{2}$ images of the vehicle license plates, respectively. Moreover, we defined the contrastive loss as the following:(5)$$L\left(w,\;{\left(\varrho_{1},\;\varrho_{2},\;\varsigma \right)}^{i}\right)=\left(1-\varsigma \right)\cdot \;L_{c}\left(E_{w}{\left(\varrho_{1},\;\varrho_{2}\right)}^{i}\right)+\varsigma L_{i}\;\left(E_{w}{\left(\varrho_{1},\;\varrho_{2}\right)}^{i}\right)$$
where (ς, $\varrho_{1}$, $\varrho_{2}$) i is the *i*th tuple of training vehicle license plates. $L_{c}$ is the partial loss function for the same license plate and $L_{c}$ is the loss function for different license plates.

## 3. Experiments Settings

### Implementation Details

We experimented the algorithm on a Core Intel i7 Machine with 64GB DRAM and NVidia Titan XP GPU, running the Ubuntu 16.04 LTS operating system. Furthermore, we developed the entire framework in Python and, most of the time, the libraries used were NumPy, matpotlib, sci-kit-learn, and SciPy.

**Training Settings:** The first experiment was performed using the Rear Vehicle Public Dataset (VRV) consisting of ten videos (i.e., 56,028 vehicles, including motorcycles and buses) with a resolution of 1902 × 1080 pixels, recorded by different cameras with a 20 min duration. Then, we set the parameters with the learning rate (γ = 0.0001) and 500 iterations. We trained the framework 80% (8 videos) and the rest (20% = 2 videos) was for the test phase. Then, in the second experiment, we used the VeRi 776 dataset, which consists of 776 vehicles with a total of 50,000 images captured by 20 different cameras. Additionally, 2 to 18 cameras with different lighting, resolution, and occlusion record each vehicle in the data sets. In both datasets, the videos are split and converted into frames that are used to extract images cropped to 96 × 96 pixels and input to the network (see Figure 3) for the DCNN architecture structure.

**Testing Settings:** We simulated the real-world problem in real-time by adding the parameters and multiplying the false negatives in the test sample set (20%). Then, we apply an appearance-based stream to both datasets independently and examine the license plate validation for a specific vehicle search. We’ve boosted the plate stream with OCR to improve the readability of plate characters under lighting variations. Then, several convolutional neural network layers were shared and feature maps were generated, which were passed to the shape stream, the plate stream, and the OCR to generate more discriminating features. The extracted feature map was further divided into local areas of the vehicle, and each area part was embedded in the pooling layer and the fully connected layers to generate descriptive feature vectors. Finally, the fully connected level of the combined stream features was passed to the attribute chaining level and softmax for matches. Additionally, we evaluated and compared the algorithm performance based on the metrics accuracy, accuracy (*P*), recall (*R*), and *F* scores. These metrics are expressed mathematically, as follows:(6)$$\mathrm{Precision}=\frac{\mathrm{TruePositives}}{\left(\mathrm{TruePositives}+\mathrm{FalsePositives}\right)}$$
(7)$$\mathrm{Recall}=\frac{\mathrm{TruePositives}}{\left(\mathrm{TruePositives}+\mathrm{FalseNegatives}\right)}$$
(8)$$F\;\mathrm{scores}=2\left[\frac{P*R}{\left(P+R\right)}\right]$$
where P and R denote precision and recall, respectively.
(9)$$\mathrm{Accuray}=\left[\frac{\mathrm{TruePositive}+\mathrm{TrueNegative}}{\mathrm{TruePositive}+\mathrm{TrueNegative}+\mathrm{FalsePositive}+\mathrm{FalseNegative}}\right]$$

## 4. Results

In this section, we present the training and validation results of our proposed algorithm (TSDCNN). We start with the performance of vehicle detection and matchings. Thus, Figure 4a–e illustrates the overall detection rate performance and effectiveness of our algorithm’s detector and vehicle appearance associations for both datasets.

In other instances, our algorithm reacted to new vehicle entries that entered and exited scenes through the camera’s left acute angle views. The detector further checked whether the entry of the new vehicle fell within the acceptance range of the trajectories. The error between the actual observation, the appearance similarity, and the predicted observation was normalized using the Euclidean method calculation. Furthermore, this confirmed the vehicle correlations and resulted in satisfactory precision, recall, and *F* score measurements during the training and validation processes. Therefore, these results are presented in Table 1 and Table 2 and visualized in Figure 5, Figure 6, Figure 7 and Figure 8. Furthermore, they are analyzed and discussed in the Results Analysis section.

## 5. Results Analysis

In this section, we analyze our TSDCNN algorithm’s results obtained from the experimental data. We trained and evaluated our detector and classifier based on a couple of combined vehicle characteristics descriptors using the VRV and VeRi776 datasets for real-time multi-vehicle tracking. The vehicle detection quality and matching are shown in Figure 4, whereas in Figure 5, Figure 6, Figure 7 and Figure 8, we illustrate the overall performance and effectiveness of our algorithm’s detector for both the training and validation phase.

The algorithm has proven effective with high performance in precision and recall, as shown in Table 1 and Table 2. This demonstrates that it could be trusted to accurately detect, learn, match and correctly classify the vehicle area of interest based on various features. However, through this process, the algorithm had challenges with the non-representative data at the beginning of the training and testing phases but gradually converged well with more training epochs. This is shown in Figure 5 and Figure 7, where the learning plots begin with difficulties in jumping up and down in statistics values due to vehicle appearance attributes and variations in entry/exit angle views in both datasets. Hence, this led to the poor recognition quality rates illustrated in Figure 4c,d and contributed to the high number of false positive classifications and mismatches, as clearly advocated in Figure 4a,b. This is highlighted in Figure 6 and Figure 8, where the algorithm’s training losses and gains at 500 and epochs project performance well on both the VRV and VeRi776 datasets. However, the algorithm showed better performance on VeRi 776 scenes, which had more difficult challenges, such as vehicle shape rotation with 225° multiple views compared to VRV datasets. As shown in Figure 4c, for the most part, the algorithm learned, identified matches, and effectively tracked all of the different vehicle shapes under various difficult conditions. This proves the robustness of our algorithms against different strong lighting and oblique viewing angles. However, despite the improved performance, Figure 5 and Figure 7 show that there were still comparable issues with the unrepresentative data, primarily during the training and testing phases for both datasets. Furthermore, it quickly converged better at epoch 270, and this showed that our algorithm had enough data during the training phase, although initially, there seemed to be problems with not getting enough data at the beginning of the training. Therefore, the bouncing [34] could be due to data fitting, as we can see that as we trained the algorithm with more epochs, we got better and more stable results for both datasets.

### Performance Comparison with State-of-the-Art Methods

To demonstrate the overall accuracy performance and specificity of our algorithms for the VRV dataset and VeRi776, we compared our precision results to state-of-the-art paradigms. The results are summarized in Table 3 and Table 4, respectively.

Therefore, from Table 3 and Table 4, it was observed that our strategy outperformed the current online state-of-the-art paradigm by 0.28% and 1.70%, respectively. This proved that the proposed approach training procedure was more convenient for real-time vehicle tracking than many other methods presented in Table 3 and Table 4. However, as illustrated in Figure 4c,d it faced challenges with angle view rotations, and as a result suffered from overlapping detection boxes, misdetection, and vehicle re-identification.

## 6. Conclusions

In this paper, we presented a deep convolutional network learning based on a three streams approach for real-time vehicle tracking and re-identification (Re-ID) problems. The study was conducted on public vehicle datasets with multiple views for vehicle detection, identification, and tracking, based on the combined three streams‘ descriptors, such as shape, plate, and OCR. This deep neural network model extracted both the global and local visual features that were more robust, representing the vehicles’ characteristics. Specifically, the stream is effectively used to extract the shape and plate. Additionally, we enhanced the license plate stream verification technique with the integration of OCR for low-resolution character reading and then used the deep neural network to calculate the similarity appearance and trajectories with the Euclidean method. This improved the overall results and proved that the proposed technique produces better detection rates and data associations. However, our algorithm experienced a poor detection rate on fast-moving vehicles. Therefore, our future work will involve implementing the algorithm for tracking multiple fast-moving vehicles on a huge dataset with a 360° angle view.

## Figures and Tables

**Figure 1 sensors-22-09274-f001:**
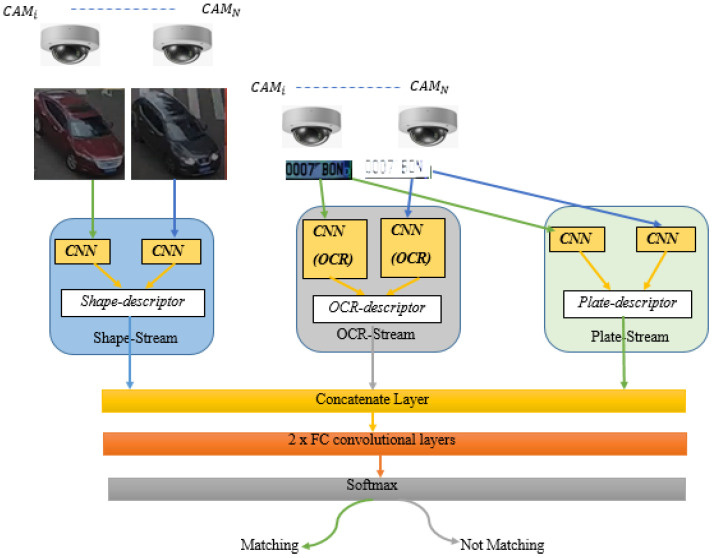
Overview of the proposed TSDCNN architecture and procedural flow.

**Figure 2 sensors-22-09274-f002:**
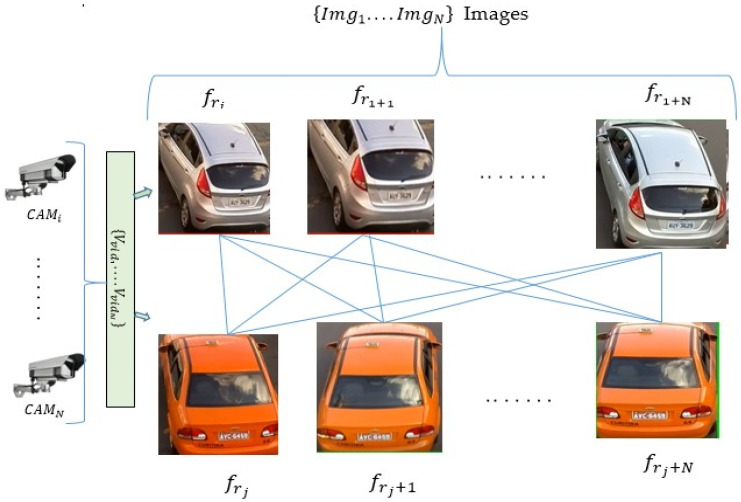
Pair-wise data augmentation for matching from input videos $\left\{V_{vid_{1}}\ldots V_{vid_{N}}\right\}$, $\left\{f_{r_{1}}\ldots f_{r_{N}}\right\}$ frames, and images $\left\{Img_{1}\ldots Img_{N}\right\}$ sequences of the distinct vehicles.

**Figure 3 sensors-22-09274-f003:**
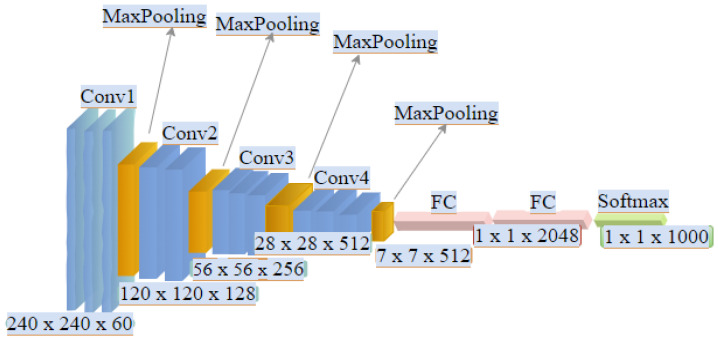
An Overview of the DCNN architectural incorporated onto the Ternion stream framework.

**Figure 4 sensors-22-09274-f004:**
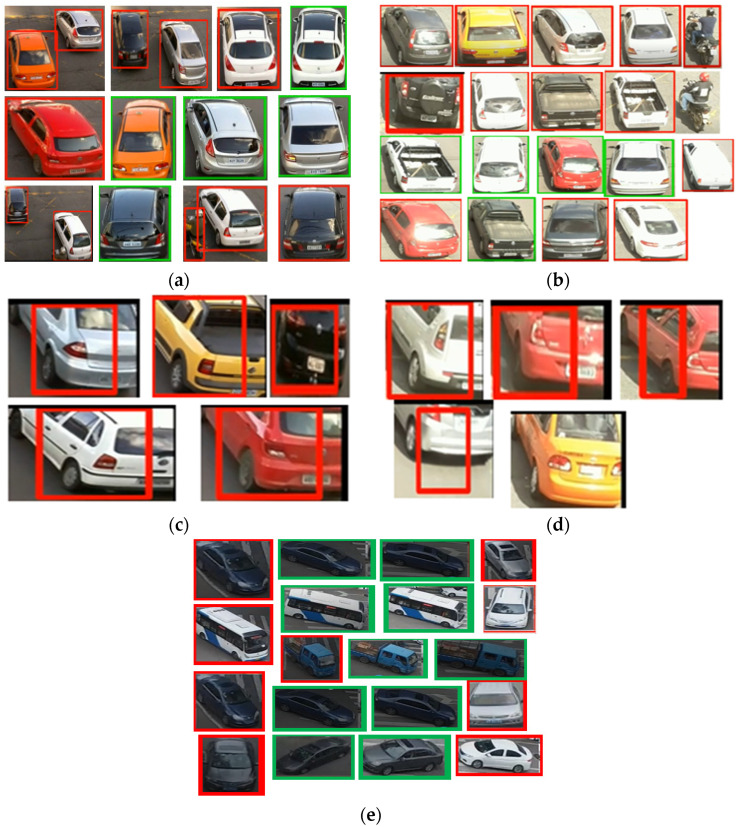
To visualize the detector and the proposed TSDCNN’s overall performance quality on vehicle matching rates across non-overlapping cameras with multiple angle views: (**a**) shows vehicles detection and matching under normal environmental conditions; (**b**) shows vehicles detection and matching under extreme illumination variations conditions. Then (**c**,**d**) are the algorithms’ poor detection quality and matching limitations under normal and extreme illumination variations conditions, respectively. Then (**e**) illustrates vehicle detection and matching under the extreme rotation variations. The red boxes around vehicles indicate the first detections, while the green boxes denotes the matches for the detected vehicle across the non-overlapping cameras.

**Figure 5 sensors-22-09274-f005:**
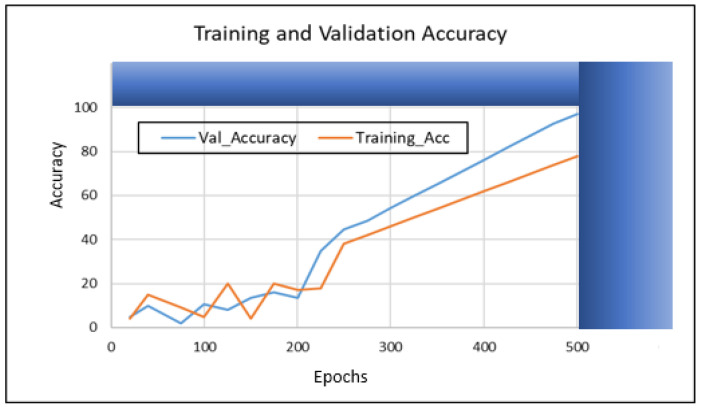
The proposed TSDCNN with 97.20% overall accuracy performance on vehicle matching rates across non-overlapping cameras with multiple angle views using the VRV dataset.

**Figure 6 sensors-22-09274-f006:**
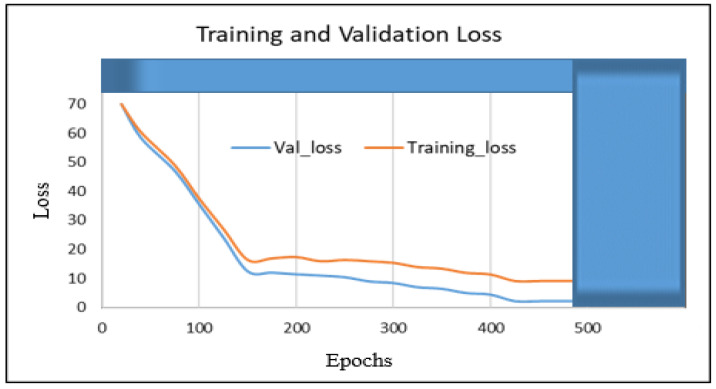
The proposed TSDCNN overall performance losses on vehicle matching rates across non-overlapping cameras with multiple angle views using the VRV dataset.

**Figure 7 sensors-22-09274-f007:**
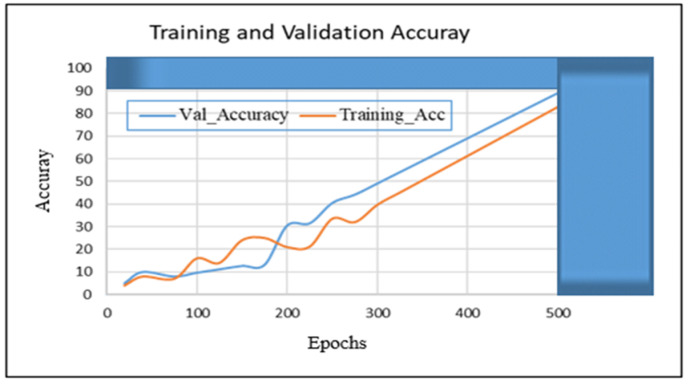
The proposed TSDCNN with 89.40% overall accuracy performance on vehicle matching rates across non-overlapping cameras with multiple angle views using the VeRi776 dataset.

**Figure 8 sensors-22-09274-f008:**
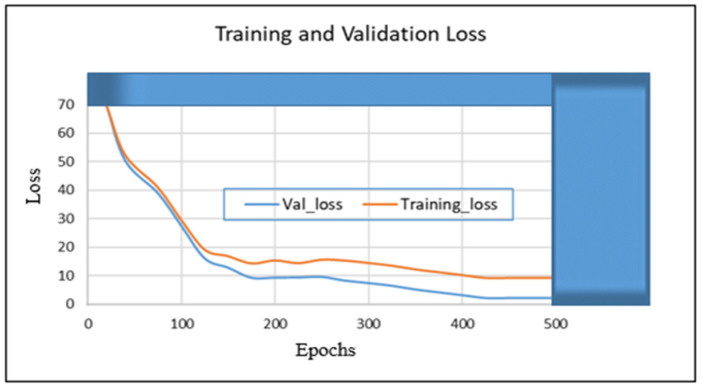
The proposed TSDCNN overall performance losses on vehicle matching rates across non-overlapping cameras with multiple angle views using the VeRi776 dataset.

**Table 1 sensors-22-09274-t001:** Performance evaluation of TSDCNN on VRV dataset.

Precision	Recall	*F* Score	Accuracy
97.20%	95.09%	96.10%	98.70%

**Table 2 sensors-22-09274-t002:** (VeRi776) Performance evaluation of TSDCNN on VeRi776 dataset.

Precision	Recall	*F* Score	Accuracy
89.40%	89.97%	90.00%	89.40%

**Table 3 sensors-22-09274-t003:** Comparison with state-of-the-art methods for testing the subset of the VRV dataset.

Methods	Precision	Recall	*F* Score	Accuracy
MatchNet [5]	98.42%	94.50%	87.10%	90.70%
LENETS [37]	97.80%	89.60%	85.20%	87.30%
MICRO [27]	97.40%	88.80%	81.80%	85.10%
VAMI [31]	98.40%	91.30%	90.60%	92.60%
**TSDCNN (Ours)**	**98.20%**	**97.20%**	**95.09%**	**96.10%**

**Table 4 sensors-22-09274-t004:** Comparison with state-of-the-art methods for testing the subset of the VeRi776 dataset.

Methods	Precision	Recall	*F* Score	Accuracy
Appearance+Color+Model [19]	87.70%	61.11%	89.27%	90.76%
FACT+Plate-SNN+STR [37]	58.21%	59.47%	61.44%	62.61%
Combining Network [27]	77.40%	60.19%	60.54%	60.60%
CNN-FT+CBL-8FT [38]	78.55%	62.62%	61.83%	60.83%
**TSDCNN (Ours)**	**89.40%**	**90.30%**	**89.97%**	**90.00%**

## Data Availability

The public datasets used can be downloaded from https://drive.google.com/open?id=0B0o1ZxGs_oVZWmtFdXpqTGl3WUU.

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
