# Peer review of "Applying Ternion Stream DCNN for Real-Time Vehicle Re-Identification and Tracking across Multiple Non-Overlapping Cameras"

_sensors, 2022, doi:10.3390/s22239274_

Round 1
Reviewer 1 Report
This paper has proposed and implemented Ternion Stream Deep Convolutional Neural Network (TSDCNN) over non-overlapping cameras and combine all key vehicle features such as shape, license plate number, and Optical Character Recognition (OCR). This paper has been written well. In order to improve the paper, the reviewer will provide several comments that should be considered carefully in this round.
-In the introduction, please add a list of contributions as points.
-Before presenting the proposed section, authors are recommended to review some related work in separate sections and then provide the background of this paper to show the system model, problem statement, etc.
-A relevant study should be reviewed as follows. CM-CPPA: Chaotic Map-Based Conditional Privacy-Preserving Authentication Scheme in 5G-Enabled Vehicular Networks.
-Please add the limitation of this paper and future work at end of the conclusion section.
-please use the LATEX (overleaf) instead of the word version.
Author Response
Reviewer 1: comments(a) In the introduction, please add a list of contributions as points.
- Authors’ Response: We proofread the whole paper, re-constructed, and re-clarified the sentences to relay the effective information. Then concretely and logically harmonized our specified research objectives with the entire Introduction at the end. This style of writing has been adopted and published by this journal (sensors) for many authors (such as: https://doi.org/10.3390/s22228819,https://doi.org/10.3390/s22228817, etc),
- We further book and send the paper to the professional English proofreading and Editing Company (Busy Bee Pty Ltd). Therefore, the typo errors, misspellings, and unclear sentences that hindered the relay of the intended information were addressed in those two phases (Our proofreading and Editing, and the professional English proofreading and editing service (The proofreading and editing certificate issued)).
(b) Before presenting the proposed section, authors are recommended to review some related work in separate sections and then provide the background of this paper to show the system model, problem statement, etc.
Authors’ Response : We reviewed literatures in the Introduction sections and logically analyzed and discussed (stated previously proposed methods' achievements (strengths) and challenges. This style of writing has been adopted and published by this journal (sensors) for many authors (such as: https://doi.org/10.3390/s22228819,https://doi.org/10.3390/s22228817, etc), hence, we are here by firmly stating that the research literature review and background, modeling and relevancy for this research are thoroughly discussed in the Introduction.
Reviewer 1: comments(c) A relevant study should be reviewed as follows. CM-CPPA: Chaotic Map-Based Conditional Privacy-Preserving Authentication Scheme in 5G-enabled Vehicular Networks.
Authors’ Response: The reviewed literatures (39 cited articles) in our paper are most relevant to our research, and the article (CM-CPPA: Chaotic Map-Based Conditional Privacy-Preserving Authentication Scheme in 5G-Enabled Vehicular Networks.) suggested to be added on the references list, it is not relevant to our research. As a result, we could not cite it.
Reviewer 1: comments(d) Please add the limitation of this paper and future work at the end of conclusion section.
Authors’ Response: We proofread the whole paper, re-constructed, and re-clarified the sentences to relay the effective information. The future work was already stated in line 455 of our conclusions section. Therefore, we now added the limitations too on this section.
Reviewer 1: comments(e) Please use the LATEX (overleaf) instead of the word version.
Authors’ Response: As far as I’m concerned we followed the Sensors Journal's Author Instructions which state and provide us with two templates (i.e: Word and LATEX). The instructions further states that authors must use either of the template provided, which is exactly what we've been doing with our previously published manuscript by choosing to use the Word template. Therefore, it surprises and sound inappropriate now when you comments on us to use the LATEX template instead of Word template for our manuscript.

Reviewer 2 Report
The following comments should be addressed before accepting this paper:
1. In the literature review part, the drawback and conflicts of current research are not highlighted. Thus, please fix this problem.
2. The contributions of this paper should be summarized by points.
3. Please give a flowchart of your proposed design
4. The following related paper is important and should be cited:
Large-Scale Bandwidth and Power Optimization for Multi-Modal Edge Intelligence Autonomous Driving
Author Response
Reviewer 2: comments(a) In the literature review part, the drawback and conflicts of current research are not highlighted. Thus, please fix this problem.
Authors’ Response: We proofread the whole paper, re-constructed, and re-clarified the sentences to relay the effective information. We further book and send the paper to the professional English proofreading and Editing Company (Busy Bee Pty Ltd). Therefore, the typo errors, misspellings, and unclear sentences that hindered the relay of the intended information were addressed in those two phases (Our proofreading and Editing, and the professional English proofreading and editing service (The proofreading and editing certificate issued)).
Then, in the Introduction sections, we thoroughly, logically analyzed and discussed (stated the currently proposed methods' achievements (strengths) and challenges).
Reviewer 2: comments(b) The contributions of this paper should be summarized by points.
Authors’ Response: We concretely and logically harmonized our specified research objectives with the entire Introduction at the end of the section. This style of writing has been adopted and published by this journal (sensors) for many authors (such as: https://doi.org/10.3390/s22228819,https://doi.org/10.3390/s22228817, etc.),
Reviewer 2: comments(c) Please give a flowchart of your proposed design.
Authors’ Response: We succinctly and logically harmonized our information flow in a picture form in Figure 1. Therefore, the Proposed architecture and flow is presented in Figure 1 of our manuscript, then followed by explanation, we believe that the design represented by the Figure 1 includes and represents the most concisely information flow for our design. If so, we therefore seek clarity as what is it missing on the current pictorial (Figure 1) information flow?
Reviewer 2: comments(d) The following related paper is important and should be cited.
Authors’ Response: We have cited 39 most relevant articles in our paper, and the article(Large-Scale Bandwidth and Power Optimization for Multi-Modal Edge Intelligence Autonomous Driving ) suggested by Reviewer 2 ,for it to be added on the references list, it is not relevant to our research. Thus, we could not cite it.

Round 2
Reviewer 2 Report
Please carefully address my comments. Otherwise, I will not recommend acceptance.
Author Response
We considered each of your comments and well addressed them accordingly. We further , illustrate that our style of writing is familiar and has been adopted by other authors who have published several articles with the journal ( sensors). In addition to address your comments, we took our script for proofreading services( the Proof-Reading Certificate is attached) for further proof read and modification of the whole paper. Therefore, we believe the comments raised were carefully addressed, please see the responses on attached pdf file.

Round 3
Reviewer 2 Report
No Comments